# Beyond Binary: (Re)Defining “Gender” for 21st Century Disaster Risk Reduction Research, Policy, and Practice

**DOI:** 10.3390/ijerph16203984

**Published:** 2019-10-18

**Authors:** Ashleigh Rushton, Lesley Gray, Justin Canty, Kevin Blanchard

**Affiliations:** 1Joint Centre for Disaster Research, Massey University, Wellington 6140, New Zealand; A.Rushton@massey.ac.nz; 2Department of Primary Health Care and General Practice, University of Otago, Wellington 6242, New Zealand; 3School of Social Sciences, University of Tasmania, Launceston, TAS 7250, Australia; Justin.Canty@utas.edu.au; 4DRR Dynamics, London WC2 2JR, UK; Kevin@drrd.org.uk

**Keywords:** gender, gender minorities, disaster, rapid review, binary

## Abstract

The dominant discourse of gender focuses on the binary of woman/man, despite the known additional risks for diverse sexualities and gender minorities in disasters. Given the small but growing body of literature concerning gender minorities in disasters, this paper sets out to explore the place of sex and gender minorities in disasters and to examine whether a binary definition needs to be extended. A five-stage rapid review was undertaken following Arksey and O’Malley’s method. Peer-reviewed journal articles in English language were sought that included disaster and gender terms in the title, abstract, and/or body of the article published between January 2015 and March 2019. The search included MEDLINE and Scopus databases. Relevant information from the studies were charted in Microsoft Excel, and results were summarized using a descriptive analytical method. In total, 729 records were identified; 248 that did not meet the inclusion criteria were excluded and 166 duplicates were removed. A total of 315 records were sourced and their full text was reviewed. Of those, only 12 journal articles included content relative to more than two genders. We also recognized that sex and gender terms were used interchangeably with no clear differentiation between the two. We recommend that disaster scholars and practitioners adopt correct terminology and expand their definition of gender beyond the binary; utilize work on gender fluidity and diversity; and apply this to disaster research, policy, and practice.

## 1. Introduction

The dominant discourse of gender within disaster focuses on the binary of woman/man or female/male [sic]. There is limited research documenting gender minorities’ experiences and health implications in disasters; what is written predominately identifies the heightened marginalization and exclusion that gender minorities face pre- and post-disaster [1,2,3]. While we acknowledge progress in gender and disaster research and practice, there is a disparity in the understanding of the lived experiences and health implications for gender minority individuals and groups. Notably, there is a gap in understanding the importance of applying appropriate gender terminology in disaster research, policy, and practice. This paper addresses this gap by providing useful instruction to minimize exclusion and marginalization of sex and gender minorities (SGM) in disaster research and management plans. This paper provides a summary of the field of gender and disaster scholarship and research which looks at the experiences of gender minorities in disasters, along with the framing of gender relating to the Western context and legislative shift. Results of a rapid review are presented, followed by a discussion of the key issues pertaining to gender minorities and disaster risk reduction (DRR), with recommendations for action.

## 2. Background

Gender and disaster scholarship grew from the recognition that women faced discrimination, exclusion, and additional challenges before, during, and after disasters. It was argued that gender played a role in how and why people were unequally affected [4,5,6,7,8,9,10,11,12,13,14]. Given that women’s mortality rates are higher [15] and domestic and sexual violence against women is increasing [4,7,16,17], there was failure to detect how gender constructs people’s experiences of disaster. An example being not all women had access to a cyclone warning in Bangladesh due to cultural restrictions; consequently, high numbers of women were injured or died during a cyclone in April 1991 [18]. It is the socio-cultural factors that contribute to women having limited access to resources, shelter, or information [19]. The predominant focus on women within gender and disaster scholarship is warranted and the research provided evidence that the social constructions of gender work to exclude and marginalize people. Given that there is specific gender and disaster research that highlights how gender influences people’s disaster experience and health outcomes, we would argue that minority genders outside of the binary must also be considered.

### 2.1. Defining Sex and Gender Minorities

For the purpose of this paper, we define gender as being the socially constructed processes and differences, often aligned with being feminine, masculine, blended elements of both, or neither. Therefore, gender minority refers to a person whose gender identity does not exclusively align with masculine or feminine polarities. Sex is defined as the physical characteristics used to identify differences between males and females; this does not mean that a person’s gender or physical sex characteristics necessarily align with their sex assigned at birth based on visible genitalia. Gender binary refers to the binary system whereby gender is assumed or considered as being a woman or man only. Therefore, gender non-binary is defined as genders outside of the woman and man binary.

### 2.2. Western Context of Gender, Sex, and Sexuality

The terms indicated in the acronym LGBTIQA+ (lesbian, gay, bisexual, transgender, intersex, queer, asexual) are derived from a Western context. It is evident in disaster responses as elsewhere that the terms are problematic in relation to conceptualizing both diverse sexualities and diverse experiences of both sex and gender in non-Western contexts alongside the existing critiques of the focus on these identity labels.

The Western concept of sex is commonly associated with physical or biological characteristics of bodies. It can be understood as equally socially constructed through both dominant discourses and biological theories, such as Laqueur [20] and Fausto-Sterling [21,22]. The social constructedness of sex categories and, specifically, the binary sex categories of male/female were illuminated by ongoing research into sex and sex characteristics. Discoveries made possible by genetics and inductive exploration of the diversity of human anatomy and experience equally disrupted the simple binary attributed to human sex differentiation, with the proposal that sex may be more properly understood as a spectrum [23]. In this regard, sex characteristics may be better described as bimodal rather than binary. As Ainsworth [23] observed, *“biologists may have been building a more nuanced view of sex, but society has yet to catch up.”* In order to perceive and redress negative effects of either omission or discrimination on the basis of sex characteristics, it is important to critically review not only the binaries of woman/man and female/male, but also of sex and gender.

In the Western context at least, gender is a concept based on the construction of systems of difference [24]. The woman/man gender binary is based on the assumed existence of only two points on a gender “scale” treated as coextensive with an apparent binary quality of physiological sex. This is commonly linked to perceptions of nature, biology, and the body and is, thus, accepted as “truth” [25]. Second-wave feminist theory contributed to disrupting an assumed link between sex and gender through arguing for a distinction between them to disrupt the equation of physical characteristics with social roles [26,27], which in itself creates another problematic binary [28]. However, conflation of physical characteristics, identity, expression, and social roles and norms remains common and has impacts on people of diverse experience and cisgender people alike.

Disrupting these dominant assumptions from a Western cultural foundation is vital to acknowledging the existence, rights, and needs of multiple genders within and beyond Western society that do not fit within the woman/man gender binary [25,29]. As such, it has relevance to efforts toward decolonization, as well as moving beyond the binary. Lorber [29] argued that gender as a paradox was a meaningless rhetoric in a binary gendered system whereby social structures continue to support gender inequality. Johnston [25] contends that it is Western culture and societal expectations that encourage and uphold the binary of gender. It is also argued that a two-gender binary frame developed into a rigid and discriminatory tool to support male privilege and power [29,30,31]. By using the woman/man binary, gender inequalities are further enforced [29]. Habitual application of the two-gender, two-sex binary frame similarly undermines queer and feminist scholars and campaigners who challenge essentialist thinking of gender and how social constructs are used as methods to reinforce “appropriate” behaviors for women and men and multiple forms of oppression such as sexism, homophobia, and racism [32,33,34]. What needs to be considered in disaster management is that neither gender nor sex are fixed terms; rather, their meanings are open to interpretation and fluidity. In this discussion, we adopt the term sex and gender minorities (SGM) from the United States (US)-based National Institutes of Health (https://www.edi.nih.gov/people/sep/lgbti/about), which is recognized as an appropriate acronym. The Yogyakarta Principles (YP) and Yogyakarta Principles plus 10 (YP+10) prepared by the International Service for Human Rights [35] provide a more useful non-binary foundation for disaster response, research and literature. This allows identification of specific dimensions of sexuality, sex, and gender that are made relevant through discrimination, omission, or persecution in the context of disaster response, independent of specific identity terms. This includes the recognition of culturally specific and intersectional experiences. The dimensions are sexual orientation, gender identity, gender expression, and sex characteristics (SOGIESC).

### 2.3. Legislative Shifts on Gender

Disasters can impact a person’s human rights. Set out in the International Covenant on Civil and Political Rights, these rights include the right to housing, the right to access healthcare, and the right to liberty and security [36]; however, enforcing humans rights is difficult. The inter-agency standing committee acts to coordinate humanitarian responses and sets out guiding principles to protect people’s human rights when affected by disaster. However, there is no binding agreement that enforces the protection of people’s human rights in crisis [36]. The guidelines are there but the willingness to use them to protect the human rights of individuals is dependent on the country(ies) involved. Given that many countries globally still persecute people whose gender and sexuality are outside the gender and heteronormative binary, it is unlikely that SGM’s human rights are going to be guaranteed protection in times of disaster.

Nevertheless, within the context of disaster, when the law, legislation, and serving government of a country do not recognize genders outside of the binary, people who do not conform to heteronormativity and who are not cisgender are marginally positioned and, thus, excluded with increased discrimination and risk. These legislation changes and challenges provide a brief look into how progress was made for issues around SOGIESC globally but also into how far the progress has yet to go. Natural hazard events disrupt society as we know it, often exacerbating inequalities and prejudices; thus, laws that protect rights and criminalize discrimination are vital to support the health and needs of those individuals who are more likely to be excluded.

There is slow progress within law to recognize genders outside of the binary. However, countries including Aotearoa New Zealand, Canada, Pakistan, Italy, Denmark, Australia, Germany, Malta, and some states in the United States of America provide a gender-neutral option on passports. Countries such as Nepal and Bangladesh provide a third gender option on their passports. Since 2015, Aotearoa New Zealand includes a “gender diverse” option when collecting statistical data, and Iceland provides a third gender option on official documents. Cuba also made significant changes to its law which now not only forbids discrimination against SGM, but also replaces the words “woman and man” with “spouse” within their definition of marriage [37]. Despite some progress globally to recognize genders outside of the binary, *“… stepping outside the bounds of heteronormativity (and by extension, gender normativity) remains illegal in many parts of the world, effectively hindering any integration of the needs of sexual and gender minorities into DRR policy and practice in a significant number of national and regional jurisdictions”* [1] (p. 22).

Countries such as Iraq, Iran, Tanzania, Burundi, Uganda, Kenya, Paraguay, Brazil, Honduras, and El Salvador are just some of the countries that still criminalize SGMs [37]. Indeed, these countries with their cultural opposition to SGM communities were often the most vocal in terms of developing political barriers related to the inclusion of SGM groups during the Sendai Framework negotiations [38]. Worryingly, the United States of America government is presently and publicly working to undo progress made by previous governments through laws that help protect the rights of SGM people including the Department of Health and Human Services Office of Women’s Health pushing to legally define gender on a biological basis [37,39]. If successful, the health of SGMs will be compromised, which could leave gender minorities with significantly poorer service provisions post-disaster.

### 2.4. Minority Genders in Disaster Risk Reduction

The Sendai Framework for Disaster Risk Reduction (2015–2030) [40] emphasizes health and the importance of providing sufficient identification of vulnerability and disaster risk in all dimensions, as well as care and training to healthcare systems and professionals [40,41,42]. Given that gender minorities may require healthcare following a disaster, it is problematic that the World Health Organization identifies gender as socially constructed identities of women and men [43] despite the recognition of third, fourth, and fifth genders in some cultures and countries [44,45] and evidence that SGMs can face restricted access to appropriate healthcare [46,47].

The United Nations office for Disaster Risk Reduction (UNDRR; formally known as UNISDR) offers “basic definitions on disaster risk reduction to promote a common understanding on the subject for use by the public, authorities, and practitioners” [48] (https://www.unisdr.org/we/inform/terminology). However, UNDRR does not include a definition of gender. Given that there is recognition that SGMs face heightened marginalization, increased discrimination, and abuse, it is disappointing that the Sendai Framework for Disaster Risk Reduction [40] does not include guidelines to minimize discrimination against diverse sexualities and gender minority groups and/or suggestions how to improve or safeguard health in relation to SGM individuals. Implementing gender analysis and providing a framework to support gender considerations in DRR is dotted throughout the document; however, disappointingly, gender within the Sendai Framework is discussed only in relation to women.

The UNDRR online registration for the 2019 Global Platform to review progress by countries on the Sendai Framework provided only two options of gender (female and male), which attributes to gender incorrectly and excludes people whose gender identity is not within the gender binary. The registration pathway sat alongside an announcement in October 2018 regarding the mainstreaming of gender throughout the 2019 global platform discussions, promoting women’s participation at international meetings [49].

A rapid review was conducted to identify peer-reviewed research journal articles in the four years since commencement of the Sendai Framework [40] to identify those with a disaster focus including minority genders, going beyond the binary. The sections below set out the approach and provide results from a rapid review of journal articles (2015 to 2019). A discussion about the findings of the review and the proposal of an appropriate definition for gender to be adopted for the disaster-related field of research are provided.

## 3. Method

A rapid review was selected to synthesize how gender is referred to and what exists in published journal articles relating to SGMs in disaster literature using systematic review methods in the time available [50]. The rapid review was undertaken following the framework of Arksey and O’Malley [51], which sets out five stages which provides a transparent and comprehensive process allowing for reliability and replication of the search strategy. This approach follows the preferred reporting items for systematic reviews and meta-analysis (PRISMA) [52].

Stage 1—Identify initial research question: This review intended to assess the extent to which disaster literature that refers to gender is limited to or goes beyond a binary framework in journal articles published in English language between 2015 and early March 2019 and to quantify the terms utilized. The timeframe was chosen to align with the commencement of the Sendai Framework for Disaster Risk Reduction [40] and the resources and timeframe for writing this paper.

Stage 2—Identify relevant studies: A wide definition of key search terms was adopted to ensure good coverage of the literature and to account for peer-reviewed journal articles that mix gender and sex terms. Inclusion and exclusion criteria took account of available time and funding limitations. We did not include hazard terms such as drought, storm surge, storms, cyclone, or typhoon, as these terms brought up many thousands of results, many of which did not appear to be disaster-related; moreover, time would not allow for a detailed review of results in the thousands. The search strategy inclusion and exclusion criteria are outlined in Table 1.

Stage 3—Study selection: Medline and Scopus searches produced 728 results in total. After exclusions and duplicates were removed, this left 315 references to be sourced and reviewed.

Stage 4—Chart the data: A summary table was completed for each of the included articles, including year, authors, and what terms were used to refer to gender.

Stage 5—Collate, summarize, and report results: The results section below provides details on the fifth stage.

The review approach involved two authors (A.R. and L.G.) with the assistance of a reference librarian (M.F.) to conduct the initial search. After duplicates were removed, an electronic library was created utilizing suitable referencing software. The two authors each searched for half of the articles, reading and assessing article content. Details were then entered into a shared Microsoft Excel spreadsheet. The results were shared between A.R. and L.G., and queries relating to the articles were discussed. The reference librarian (M.F.) assisted with sourcing full text of articles that the two authors were unable to locate initially.

## 4. Results

The initial search identified 729 records; of these, 248 records did not meet the inclusion criteria and were excluded, and 166 duplicates were also removed. This left 315 records to be sourced and reviewed as full texts. Full texts were located, obtained, and reviewed. Once full texts were reviewed, a further 55 records were excluded. Two records were pre-2015, two were conference presentation abstracts, and one was a news summary. Four records were not related to disasters, and one specifically looked at volunteers in humanitarian settings. A small number of articles were about dental forensics and body identification methods (*n* = 5), health skill development around procedures (*n* = 8), and dietary intake (*n* = 1). Results relating to books or book chapters were excluded (*n* = 11). Those related to school, children, youth, and/or adolescents were excluded (*n* = 15), and we observed that none of these included content relating to more than two genders. A further five articles were excluded as they did not include any reference to gender or sex. This left 260 journal articles.

Two articles meeting inclusion criteria had content relating to gay men. Whilst not having specific content relating to more than two genders, these articles, along with one article that briefly noted transgender groups, did acknowledge non-gender-normative and heteronormative identities and included important considerations in relation to disaster risk reduction and the aftermath following disaster (see Appendix A). When reviewing the full-text articles for the purpose of this review, we determined that “LGB” was related to sexuality and not gender; however, we included papers that used the acronym LGBT, given the fluidity of “T” (i.e., transgender and/or transsexual individuals), which may identify as a gender beyond the binary [53]. During the review, we recognized that 200 out of the 260 articles used the terms sex and gender interchangeably with no clear distinction between the two terms. Figure 1 shows the record selection process.

Following our review of 260 peer-reviewed journal articles, we noted that only 12 went beyond the gender binary of woman/man. Two out of the 12 articles referred to the acronym LGBT(I/Q) and a further one article mentioned transgender groups. Four articles reviewed included the terms gender minorities and non-binary. Unsurprisingly, only five articles that we reviewed discussed specific genders beyond the binary such as Aravani, Fa’afafine, Baklâ, and Waria. Another observation is that eight of the 12 articles specifically focused on gender minorities and/or LGBTI groups. A further two articles specifically focused on gay individuals; however, as both papers included the acronym LGBTI/LGBTQ, they were included. The remaining two articles did not specifically aim to examine minority groups in DRR, yet they made reference to genders outside of the binary. Table 2 below provides a summary of these papers.

## 5. Discussion and Conclusions

The aim of this paper was to explore the place of gender in disaster scholarship and to examine the persistent use of a binary definition despite scientific and social shifts to recognize SGM experiences beyond this binary. The rapid review identified how gender was referred to within the academic literature on disaster. Our initial literature search of the experiences of gender minorities in disaster identified three things. Firstly, there is limited literature relating to gender minorities and disaster. Secondly, the papers that did discuss gender and sexual minorities discussed the health impacts on and discrimination against gender minorities in disaster events. Thirdly, we identified that gender and sex were used interchangeably with female, male, women, and men, and there was no clear distinction between the two. The scope of this paper was not able to ask or identify *why* this may be, but suggested reasons are discussed. We also make a recommendation for further work in this area.

Through the rapid review, we identified only 12 journal articles that considered minority genders outside of the binary in relation to disasters; this accounted for only 4.61% of the 260 included papers. It is fair to say that, of the 12 papers that considered more than two genders, eight papers set out to intentionally explore minority genders. The fact that the majority of papers reviewed adhere to the gender binary could be due to gender-normative assumptions, as empirical data more commonly only include woman/man or female/male. The majority of full-text articles reviewed referred to the binary woman/man. This may reflect the demographics of participants, or it might be a reflection of limited options provided to research participants. Additionally, there may be a lack of understanding from researchers as to possible genders outside of the binary. For example, one article looking at the use of social media in the Philippines to recruit participants in a post-disaster setting claimed the demographics of their study sample were similar to the country demographics in relation to gender, age distribution, and level of education. However, there was no mention of non-binary genders such as baklâ in this paper. Another suggestion as to why disaster research and policy tend to neglect gender minorities could be that scholars and practitioners are not aware of genders outside of the binary, or that they may wish to enforce the binary due to their own personal beliefs. We make these suggestions for further consideration.

We also cannot ignore that in some countries, gender minorities may be stigmatized and put at risk for publically identifying outside of the gender binary; thus, we must consider that some researchers may consciously avoid including gender minorities in their work in order to avoid drawing attention to those individuals and communities, thereby averting putting them in danger. In addition, sex and gender minorities may also work to avoid disclosure to minimize the risk of discrimination, violence, or persecution.

Unfortunately, due to the limitation of this paper, we cannot explore the reasons behind *why* there are few articles that include gender minorities in more detail, but it is clear that the following question lies here: How do we, as disaster scholars, ethically and appropriately include gender minorities in disaster research? This question, we hope, starts a conversation and challenges the thinking in disaster research, policy, and practice around gender.

The role of research literature and the media in disasters contributes to the (re)shaping and (re)telling of people’s disaster narratives. However, commonly, the stories and narratives of SGM individuals are omitted or ignored in disaster; they become invisible and excluded from the public’s construction and understanding of particular natural hazard events [58]. This exclusion contributes to minority groups’ vulnerability and lack of voice, but also ignores their resilient capacities, strengths, and contributions in the wider community. Complying with the gender binary, therefore, excludes non-binary minority genders from disaster discourse. It is imperative that gender minority voices and experiences become included in the wider discussions on disaster risk reduction.

Disasters provide an opportunity to explore vulnerabilities and resilience further when, sometimes, the private is forced into the public. Disaster events can assist in developing understanding on not only how natural hazards can impact populations, but how we as scholars and practitioners can work *with* communities to minimize risk for *all* in the face of natural hazards and the changing climate. There is a small body of literature that discusses people who are lesbian, gay, bisexual, transgender, intersex, queer, androgynous, and other genders in relation to experiences of disaster events. Dominey-Howes et al. [3] described how intersex, transgender, lesbian, and bisexual women were at risk of “corrective rape” following the Haiti earthquake. Following Hurricane Katrina, same-sex couples were not acknowledged as being in a relationship and, therefore, were excluded from receiving assistance [3,64], as the disaster model at the time of Hurricane Katrina was based on heteronormative assumptions of relationships [64,65].

Current DRR frameworks and policies further exacerbate disadvantages for sexual and gender minorities in disaster [3]. Gender minorities have been denied access to aid and emergency accommodation [2], which includes the arrest of Sharli’e Dominique, a transgender women for showering in a women’s restroom post Hurricane Katrina [65]. During the Mt Merapi eruption in 2010, waria were faced with no safe accommodation options when registering at a shelter given that registration only had gender binary options; in addition, no information was collected on waria when they evacuated [66].

People who identify outside of the normative gender binary of woman/man and who are not heterosexual or cisgender can live in fear of harassment, abuse, and violence, which can be heightened in disasters [2,3,53,66]. There were concerns from individuals seeking food parcels from St. Vincent de Paul because of the religious association of the charity, and the emergency personnel assuming cisgender identity of people affected by the Christchurch earthquake [55]. Similarly, the Federation Emergency Management Agency (FEMA) followed *“a heteronormative system of binary terminology and legislative inflexibility”* [64] (p. 7), thereby excluding people within the New Orleans population that did not fit within the category of heterosexual and cisgender. Health is a major concern for individuals and groups who do not conform to heteronormative cultural norms. Gay men in Kingston, Jamaica known as “The Gully Queens” are discriminated against and forced to live out on the banks of a gully given that there is no safe shelter; the individuals who live on the river banks are at high risk of being dragged away by the water during hurricane and flood events [67].

The examples provided highlight how exclusion of diverse sexual and gender minorities is *“underpinned by heteronormative assumptions in disaster response and recovery”* [3] (p. 914). The woman/man binary and terminology is inadequate to address the increased marginalization of sex and gender minorities in disasters coupled with the Western perspective asserting gender dichotomy [1]. The examples discussed illustrate the importance of using appropriate and correct terminology in disaster management and for governments and non-governmental organizations (NGOs) to *“ensure safe access to and experiences in accommodation for all social groups”* [2] (p. 250).

Gender construction that represents binary oppositions, derived from the dominant Western binary discourses, *“problematizes claims to the universal applicability of the concepts around sex and gender”* [24] (p. 84). This is seen in recent UNDRR policy documents and guidelines, as well as registration of the leading global DRR meeting for DRR policy and action. This not only upholds gender inequality, thereby excluding gender minorities, but also contributes to preparedness, response, and recovery efforts that may not be as effective in reaching individuals and communities that are at greater risk and in need of assistance [2].

Health is a key element of the Sendai Framework for Disaster Risk Reduction (2015–2030) [40,41,42], yet there is little discussion in the disaster literature reviewed of the importance of providing gender-sensitive appropriate healthcare in relation to SGM individuals by health professionals who are sensitive and aware of the historical marginalization and persecution of SGM groups [46]. Healthcare professionals and organizations assisting in post-disaster events should have knowledge of some of the barriers, along with specific health risk factors that some people within SGM communities may face [46]. Health consideration and physical safety is paramount pre- and post-disaster given that people may require specialist care from injury or may need assistance with on-going health issues that may be exacerbated by disaster.

## 6. Recommendations

It is evident from our review that, firstly, there is a dearth of research on gender minorities in disaster events. Disaster research that assesses gender in some form need to acknowledge genders outside of the binary; this will not only make research more comprehensive in terms of representing more people within their study, but it may also work to undo some of the inequalities in disaster research and practice. Research should also not only look at the challenges people who identify outside of the gender binary may face, but also look at their capacities, strengths, and contributions to DRR.

Researchers and practitioners need to utilize the growing body of work on gender beyond the familiar binary, diversity of sex characteristics, and the issues related to imposing overly restrictive categories on diverse experiences [68] and apply it to disaster scholarship and policy. A simple and effective starting point will be for the UNDRR to adopt the definitions of sexual orientation, gender identity, gender expression, and sex characteristics captured in the preambles of YP and YP+10 as a guide for future disaster research and policy. Using an expanded definition of gender and sex, we hope, will contribute to minimizing the exclusion of SGM perspectives in disaster policy, practice, and research. A further step would be to move “*beyond the binary*” and consider not only where sex and gender are relevant considerations but also to disrupt automatic assumptions that they are omnirelevant categories.

The scope of this paper was not able to ask or identify *why* gender and sex terms were used interchangeably. We propose that research can be conducted to identify why and how these choices influence policies and practices in DRR. We also posit that, for some articles, this may have been related to limited knowledge of the differences, which is of concern. We, therefore, urge researchers in the field of DRR to familiarize with and better understand the sex and gender terms within the wider DRR terminology work happening at both national and international levels. We further recommend that suitably skilled researchers look at papers in non-English languages to add to this review, especially in relation to non-Western approaches.

Finally we advocate for DRR researchers and practitioners to utilize the growing body of work on gender diversity and issues concerning the gender binary [68], incorporating indigenous studies and scholarship on decolonizing gender identities, expressions, and experiences, as well as the scientific literature on more nuanced views on sex and sex diversity [23]. Within this, researchers must address methodological and safety concerns relevant to SGM for the countries and cultures under consideration and recognize where these may constitute limitations to participation and findings. They should also consider the gender identities, gender expressions, and people with diverse sex characteristics represented in the country of the research focus, and ensure that, if gender is included as part of the research, that all genders appropriate to that country and/or culture are recognized and included.

**Limitations:** The quality of evidence in primary research reports was not appraised and was limited to reports and documents published in English language. The review was not a full systematic review, and we limited the search terms on types of hazards due to the high volume of results produced for review in the time available.

## Figures and Tables

**Figure 1 ijerph-16-03984-f001:**
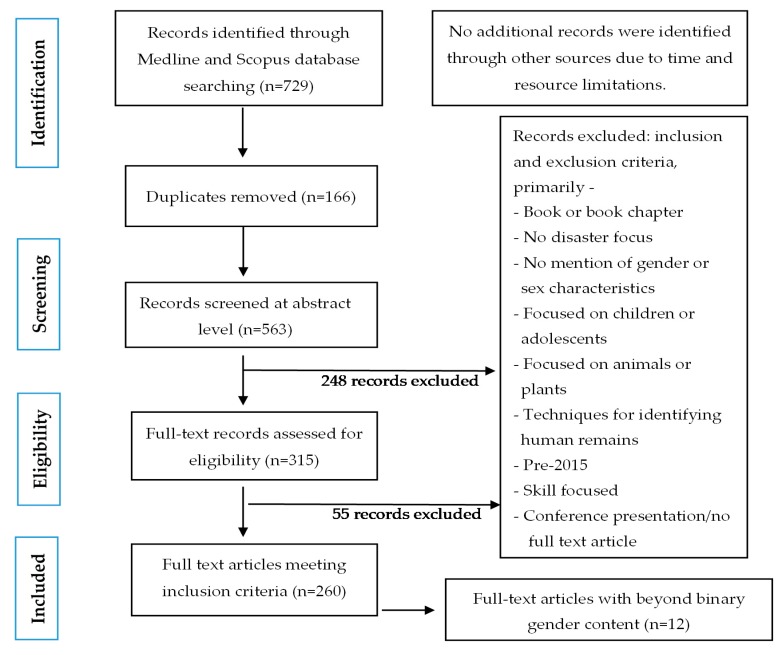
Preferred reporting items for systematic reviews and meta-analysis (PRISMA) flow diagram for rapid review article selection.

**Table 1 ijerph-16-03984-t001:** Search strategy inclusion and exclusion criteria.

**Search Strategy**		
Male, men, men’s, man, man’s		
Female, women, women’s, woman, woman’s		
Gender or gender with identit or nonconforming, non-conforming, divers, binary, nonbinary, other, unspecified, nonspecified, non-specified, androgyn, intermediate, minorit, spectrum, cisgender, transgender, trans-gender, transsexual, trans-sexual, queer, lgbt, glbt, twospirit, two spirit, two-spirit, uranian, fa’afafine, faafafine, berdache, pangender, bigender, genderqueer, androgyne, intergender, intersex, third sex, fourth sex, ashtime, mashoga, mangaiko, palao’ana, palaoana, fakaleiti, mahu wahine, mahu vahine, whakawahine, akava’ine, akavaine, waria, warias, baklâ, baklas, binabae, bayot, bantut, bading, femminiello, muxe, biza’ah, bizaah, travesties, xanith, khanith, tritiya-prakrti, ubhatobyanjanaka, pandaka, quariwarmi		
Disaster, earthquake, avalanche, landslide, mudslide, land slide, mud slide, tsunami, tornado, mass casualty, volcan, flood.		
**Inclusion and Exclusion Criteria**		
Included	Included	Excluded
Year	2015 to March 2019	Pre-2015 and after March 2019
Language	English language articles	Non-English language articles
Format	Journal articles	Book or book chapter
Disaster	Disaster focus	No disaster focus
Gender terms		Articles that did not mention gender or sex terms
Age		Articles focused on infants or children (age 10 and under)
Non-human	Adults	Animals and plants
Search terms within articles		
Gender (binary/non-binary), female, male, woman/women, man/men, other gender/sex terms		

**Table 2 ijerph-16-03984-t002:** Rapid review: papers referring to minority genders outside of the binary (*n* = 12).

Year	Citation	Beyond Binary Summary
2018	Myers, A.; Sami, S.; Onyango, M.A.; Karki, H.; Anggraini, R.; Krause, S. Facilitators and barriers in implementing the Minimum Initial Services Package (MISP) for reproductive health in Nepal post-earthquake. *Confl. Health* **2018**, *12*, 35. [54]	LGBTI
2018	Gorman-Murray. A.; McKinnon, S.; Dominey-Howes, D.; Nash, C.J.; Bolton, R. Listening and learning: Giving voice to trans experiences of disasters. *Gend. Place Cult*. **2018**, *25*, 166–187. [55]	Gender minorities Transgender Baklâ Waria Aravani
2018	Dominey-Howes, D.; Gorman-Murray, A.; McKinnon, S. On the disaster experiences of sexual and gender (LGBTI) minorities: Insights to support inclusive disaster risk reduction policy and practice. *Aust. J. Emerg. Manag*. **2018**, *3*, 60–68. [53]	Gender minorities Non-binary Sexual minorities LGBTI
2017	Wisner, B.; Berger, G.; Gaillard, J.C. We’ve seen the future, and it’s very diverse: Beyond gender and disaster in West Hollywood, California. *Gend. Place Cult*. **2017**, *24*, 27–36. [56]	Non-binary Cisgender, Transgender Transsexual Gender fluid
2017	Ong, J.C. Queer cosmopolitanism in the disaster zone: ‘My Grindr became the United Nations’. *Int. Commun. Gaz*. **2017**, *79*, 656–673. [57]	LGBTQ
2017	McKinnon, S.; Gorman-Murray, A.; Dominey-Howes, D. Disasters, queer narratives, and the news: How are LGBTI disaster experiences reported by the mainstream and LGBTI media? *J. Homosex*. **2017**, *64*, 122–144. [58]	Cisgender Transgender Gender minorities
2017	Gaillard, J.C.; Gorman-Murray, A.; Fordham, M. Sexual and gender minorities in disaster. *Gend. Place Cult*. **2017**, *24*, 18–26. [1]	Gender and sexual minorities Baklâ Aravani
2017	Gaillard, J.C.; Sanz, K.; Balgos, B.C.; Dalisay, S.N.; Gorman-Murray, A.; Smith, F.; Toelupe, V.A. Beyond men and women: A critical perspective on gender and disaster. *Disasters* **2017**, *41*, 429–447. [59]	Aravani Fakaleiti Mahu Whakawahine Baklâ Waria Fa’afafine
2017	McKinnon, S.; Gorman-Murray, A.; Dominey-Howes, D. Remembering an epidemic during a disaster: Memories of HIV/AIDS, gay male identities and the experience of recent disasters in Australia and New Zealand. *Gend. Place Cult*. **2017**, *24*, 52–63. [60]	LGBT Gender minority groups
2017	Yamashita. A.; Gomez, C.; Dombroski, K. Segregation, exclusion and LGBT people in disaster impacted areas: Experiences from the Higashinihon Dai-Shinsai (Great East-Japan Disaster). *Gend. Place Cult*. **2017**, *24*, 64–71. [61]	LGBT Transgender Waria
2015	Işık, Ö.; Özer, N.; Sayın, N.; Mishal, A.; Gündoğdu, O.; Özçep, F. Are women in Turkey both risks and resources in disaster management? *Int. J. Environ. Res. Public Health* **2015**, *12*, 5758–5774. [62]	Transgender groups
2015	McSherry, A.; Manalastas, E.J.; Gaillard, J.C.; Dalisay, S.N. From deviant to bakla, strong to stronger: Mainstreaming sexual and gender minorities into disaster risk reduction in the Philippines. *Forum Dev. Stud*. **2015**, *42*, 27–40. [63]	Sex and gender minorities Baklâ

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
