# Peer review of "Beyond Binary: (Re)Defining “Gender” for 21st Century Disaster Risk Reduction Research, Policy, and Practice"

_ijerph, 2019, doi:10.3390/ijerph16203984_

Round 1

Reviewer 1 Report

The experience of trans and gender non-binary individuals during disasters is important and this work is timely.  Unfortunately, three major issues limit this manuscript's ability to be published in its current form.

1) The introduction is too long and not focused enough on the topic of the paper.  It reads more like a dissertation than a journal article.  Pare down the theory and focus succinctly on the issue of gender and disaster research.

2) Choose consistent and up-to-date terminology and use these terms throughout the paper.  "Homosexuality" and "homosexuals" are outdated and stigmatizing terms.  "SOGIESL people" does not make sense grammatically.  NIH, in the US, uses "sexual and gender minority (SGM)" - other terms are fine, as long as they are applied consistently and sensibly.

3) There is no discussion of the results of the rapid review beyond a table.  Contextualize these findings.  Spend less time on the introduction and more time on the results and discussion to give readers a takeaway message from the most recent disaster science.

Author Response

Thank you for your review, please find attached our response to your comments.  We feel the reviewer comments have guided the significant improvement of this manuscript. 

Reviewer 2 Report

The premise of the article is very simple, ‘we’, as in gender academics, have moved past understandings of ‘gender’ as a binary concept, to accepting it as a fluid notion. The acceptance of diversity is also reflected in the UN's Yogyakarta Principles, for example, which suggests a non-binary understanding of gender has become an institutional norm. if this is the case then we might expect to see this reflected in research – here in the ‘disaster’ context – and the study explores if this is the case. Not surprisingly it finds that it is not the case and then recommends it should be.

The method deployed is to undertake a rapid review of relevant articles over a specific time frame. Of the 260 articles that met the inclusion criteria only 12 use ‘beyond binary’ understandings of gender. It lists these.

The paper in terms of content is mostly ‘background’ with the ‘discussion and conclusions’ consisting of only one page, with a paragraph about health rather added on.

I have no problem with what is here, or the argument, I just feel it is rather light on substance. For those who do not ‘get’ why we need to go beyond binaries, this will not enlighten them, it may actually make them think, ‘well if no one else is doing it, why should I?’ For those who already think in non-binary terms, this offers little new other than one stat they could use – of 260 articles only 12 were non-binary. Academically speaking it is then weak in terms of argument and as advancing thinking.

The rapid review findings need to be better contextualised. Why do researchers still not go ‘beyond-binaries’? is it because they don’t know any better? Is it because they agree with binaries? Is it because they are working in contexts when ‘minorities’ keep themselves hidden?  In many of the examples the authors use the ‘minority’ groups mentioned are very specific, named and known and should be ‘counted’. This is not the case in many countries and cultures, where seeking individuals out based on their difference and effectively then highlighting this difference, would actually put those people in danger. So how to do non-binary research ethically in contexts where being non-binary beings is dangerous and thus is often kept hidden from view?this would be a useful discussion.

Does the disaster context, where often what is hidden / private becomes visible / public, provide a special window of opportunity to apply a non-binary lens?  If this is the argument, which is could be, then make it so. If it is not, then what is the argument?

The paper notes  ‘gender’ and ‘sex’ were used interchangeably in many of the articles – perhaps then construct a typology, a spectrum of ‘gender’ usage rather than you yourselves adopt a binary approach in the analysis. Again, this would make for a more interesting argument since at present the findings are unsurprising and the paper then adds little to our knowledge of going ‘beyond-binary’ or of disasters, which is a shame since it is an important issue to discuss.

Author Response

Thank you for your review.  We feel the reviewer comments have contributed to significantly improving this manuscript.

Reviewer 3 Report

I found this a fascinating article, generally of a high standard, which I thoroughly enjoyed reading. In particular, the clear definition of terms in 2:1 was helpful. The search strategy was clear and pragmatic and the authors do not overclaim their findings.

I would be interested in some elaboration as to the inclusion of articles about those who identify as homosexual and do not apparently identify in any way as non-binary or queer. Given the shortage of relevant articles, I can understand why these might have been included but would suggest that these are not directly relevant to the topic.

I would also like to understand why in the table that includes search terms, possessive terms (man's, men's) are included but the equivalent 'woman's / women's) - or have they just been accidentally omitted from the table?

I would suggest that the final sentences need reframing. The last sentence is not currently a stand alone sentence. It could, for example, be prefaced with 'They should also.....' It would be a shame to end the article with a lack of clarity.

I appreciate that the combination of knowing one's own work and seeing what one thinks one has written along with auto correct functions result in typos. There follows a list of errors that I noticed whilst reading that the authors might wish to look at. Please see these as a wish to be helpful rather than to be critical!

Abstract L.26 Omit 'to'

P1 L34 add a semi-colon after 'disasters'

P2 L 13 'peoples'' rather than 'peoples' (apostrophe added as it is a possessive). This occurs again P4 L41; P4 L42; P4 L45; P9 L25. Likewise, P3 L6 individuals'; P9 L26 public's (not publics);

P2 L17 either 'a cyclone warning' or 'cyclone warnings'.

P2 L38 'who' not 'whom'

P2 LL39-42 this point would be clearer if broken into shorter sentences

P2 L46 do you mean 'personnel' rather than 'personal'?

P3 L4 a semi-colon would be clearer than a comma

P4 L20 delete 'it'

P4 L21 suggest 'with' rather than 'to'

P4 L31 suggest lower case 't' for 'the'

P4 L33 'Services' does not need an apostrophe

P4 L36 Do you mean 'health' or do you mean 'service provision'?

P4 L44 Omit 'against'

P4 L45 suggest 'heteronormal' rather 'heteronormativity'

P4 L47 replace 'whether' with 'if' or 'when'

P5 L4 suggest 'be' rather than 'get'

P5 L6 suggest 'emphasises' rather than 'gives emphasis'

P5 L14 does this quotation need a page number?

P5 L15 'does' rather than 'do'

P5 L23 suggest add 'only'

P5 L32 add 'the' before Sendai framework

P8 L1 'peer reviewed' not 'peer review'

P8 L4 'unsurprisingly' not 'unsurprising'

P9 L13 suggest insert 'that' after 'The fact'

P9 L24 'contributes' rather than 'contribute'

P9 L27 'contributes' rather than 'contributions'

P9 L43 suggest 'of' not 'to'

P9 L49 'needs' rather than 'need'

P9 L50 suggest a semi-colon rather than a comma

P10 L1 insert 'of' before 'the inequalities'

P10 L14 suggest the removal of inverted commas from 'why' to add clarity

P10 L15 suggest rather than 'effect'

P10 LL25-28 Suggest that 'country' does not need an upper case 'c'

Author Response

Thank you for your review.  We feel the reviewer comments have significantly contributed to a much improved manuscript.  Our responses are attached.

Round 2

Reviewer 1 Report

This paper is much improved.  I appreciate the shift of content into the Discussion section.  My only minor suggestion is to reorganize the section on "Legislative Shifts on Gender."  I would advocate for placing paragraphs 3 and 4 (beginning with "Disasters can impact...") before paragraphs 1 and 2, to make the point that legal recognition of gender is important in the context of disasters, and that some countries do this better than others.

Author Response

We sincerely thank Reviewer One for their further consideration of our manuscript.  We agree with Reviewer One that by moving paragraphs in section 2.3 this has further improved the flow and emphasis of this section.  Thank you.  We have re-numbered references 36-39 as they now appear.